Cropping system diversification for food production in Mindanao rubber plantations: a rice cultivar mixture and rice intercropped with mungbean

Hondrade Rosa Fe 1
Hondrade Edwin 1
Zheng Lianqing 2 3 11
Elazegui Francisco 4
Duque Jo-Anne Lynne Joy E. 5
Mundt Christopher C. 6
Vera Cruz Casiana M. 7
Garrett Karen A. karengarrett@ufl.edu 2 8 9 10
1 University of Southern Mindanao , Cotabato , Philippines
2 Department of Plant Pathology, Kansas State University , Manhattan , KS , United States
3 Department of Statistics, Kansas State University , Manhattan , KS , United States
4 International Rice Research Institute , Los Banos , Philippines
5 University of Southern Mindanao , Kabacan , North Cotabato , Philippines
6 Plant Pathology Department, Oregon State University , Corvallis , OR , United States
7 Genetics and Biotechnology Division, International Rice Research Institute , Los Banos , Philippines
8 Institute for Sustainable Food Systems, University of Florida , Gainesville, FL , United States
9 Plant Pathology Department, University of Florida , Gainesville, FL , United States
10 Emerging Pathogens Institute, University of Florida , Gainesville, FL , United States
11 Current affiliation:  Gilead Sciences, Inc. , Foster City, CA , United States
Wratten Stephen
Electronic publication date: 2017 Feb 8
Publication date: 2017
Volume: 5
Electronic Location ID: e2975
Received 2016 Oct 6; Accepted 2017 Jan 10
Copyright: ©2017 Hondrade et al.
Copyright year: 2017
Copyright holder: Hondrade et al.
License: This is an open access article distributed under the terms of the Creative Commons Attribution License, which permits unrestricted use, distribution, reproduction and adaptation in any medium and for any purpose provided that it is properly attributed. For attribution, the original author(s), title, publication source (PeerJ) and either DOI or URL of the article must be cited.
License URL: https://creativecommons.org/licenses/by/4.0/

Keywords: Agricultural diversification, Agroforestry, Cultivar mixtures, Hevea brasiliensis, Intercropping, Mindanao, Mungbean, Philippines, Rice, Rubber plantations

Funding: US Agency for International Development Award No. EPP-A-00-04-00016-00 Kansas Agricultural Experiment Station Contribution No. 15-332-J University of Florida Asian Development Bank This work was supported by the US Agency for International Development through a grant to the IPM CRSP at Virginia Tech (Award No. EPP-A-00-04-00016-00), as part of a project funded for Southeast Asia to M Hammig, M Shepard, and G Carner of Clemson, by the Kansas Agricultural Experiment Station (Contribution No. 15-332-J), and by the University of Florida. Additional support was provided by the Asian Development Bank to the International Rice Research Institute through the Consortium for Unfavorable Rice Environments. The funders had no role in study design, data collection and analysis, decision to publish, or preparation of the manuscript.

==============================
Including food production in non-food systems, such as rubber plantations and biofuel or bioenergy crops, may contribute to household food security. We evaluated the potential for planting rice, mungbean, rice cultivar mixtures, and rice intercropped with mungbean in young rubber plantations in experiments in the Arakan Valley of Mindanao in the Philippines. Rice mixtures consisted of two- or three-row strips of cultivar Dinorado, a cultivar with higher value but lower yield, and high-yielding cultivar UPL Ri-5. Rice and mungbean intercropping treatments consisted of different combinations of two- or three-row strips of rice and mungbean. We used generalized linear mixed models to evaluate the yield of each crop alone and in the mixture or intercropping treatments. We also evaluated a land equivalent ratio for yield, along with weed biomass (where Ageratum conyzoides was particularly abundant), the severity of disease caused by Magnaporthe oryzae and Cochliobolus miyabeanus, and rice bug (Leptocorisa acuta) abundance. We analyzed the yield ranking of each cropping system across site-year combinations to determine mean relative performance and yield stability. When weighted by their relative economic value, UPL Ri-5 had the highest mean performance, but with decreasing performance in low-yielding environments. A rice and mungbean intercropping system had the second highest performance, tied with high-value Dinorado but without decreasing relative performance in low-yielding environments. Rice and mungbean intercropped with rubber have been adopted by farmers in the Arakan Valley.

Introduction

The spread of agricultural non-food systems, such as rubber plantations, is a major factor for smallholder farmers and farm laborers, who often must navigate potential shifts from traditional swidden systems (Fox & Castella, 2013; Josol & Montefrio, 2013; Li et al., 2014; Manivong & Cramb, 2008; Mertz et al., 2013; Montefrio & Sonnenfeld, 2013; Van Vliet et al., 2012; Vongvisouk et al., 2014). Increasing rubber production has the potential to reduce household food security, if less land is used for local food production (Weinberger, 2013). In addition, the cleared land between young trees is wasted from an economic standpoint, and may be subject to erosion, if it is not planted with other crops. The use of intercropping of food crops in tree plantations has the potential to address both of these problems. Rubber trees may be intercropped with a range of other plant species, including food crops and tea, cocoa, coffee, rattan, fruit trees, and cinnamon (Jessy, Joseph & George, in press; Pathiratna & Perera, 2006; Penot & Ollivier, 2009; Wu, Liu & Chen, 2016). Guo et al. (2006) concluded that rubber-tea intercropping provided an economic benefit compared to separate rubber and tea monocultures in Hainan, China. Rubber intercropping with a crop like banana, before latex is produced, may even ultimately improve rubber production (Rodrigo et al., 2005). Systems of rubber intercropping in Nigeria often gave improved productivity, including systems of intercropping with soybean and melon, or with melon and maize (Esekhade et al., 2003). In Kerala, India, adoption of intercropping of rubber with pineapple, banana, and cassava was reported as most common (Rajasekharan & Veeraputhran, 2002).

Planned diversification of plantations can be implemented at multiple scales. Smallholder farms have included both small-scale rubber production and rice (Dove, 1993). An annual cropping system intercropped in rubber plantations might include intercropping of a staple cereal with a legume for additional nutritional benefits to local consumers. Including a legume in the system can also increase N availability in soils (Schroth, Salazar & Da Silva, 2001; Van Noordwijk, Cadisch & Ong, 2004). The use of cultivar mixtures may provide additional benefits (Finckh et al., 2000; Garrett & Mundt, 1999; Meung et al., 2003). A striking example is the great success of rice mixtures composed of a rice cultivar with higher economic value but susceptible to rice blast, with a rice cultivar with lower economic value but resistant to rice blast (Zhu et al., 2000). Diversification to include plants that support the natural enemies of crop pests (Gurr et al., 2016) or that repel pests or attract them away from crops (Khan et al., 2000) may also provide benefits to production. The effects of increased system diversity can be difficult to predict, however, and functional diversity designed to achieve particular cropping system goals may be more useful than haphazardly constructed diversification. The effects of system diversification need to be studied in field experiments because what seem like intuitive outcomes may not be observed in practice.

In the Arakan Valley of Mindanao, the southernmost island group of the Philippines, demand for rubber drove replanting because the majority of rubber trees in the region were more than 30 years old and had reduced latex production (RF Hondrade, pers. obs., 2006). When replanting, farmers would need to wait approximately five to seven years before they could begin tapping the new rubber trees. This interval is an opportunity for the production of annual crops such as upland rice, corn, vegetables, and grain legumes between the immature rubber trees, before the rubber trees are large enough that interspecific competition becomes an important factor. Such intercropping can produce household food and may also generate income while waiting for the rubber trees to reach a productive age.

The objectives of this study were to evaluate the effects of three types of diversification in rubber agroforestry that included mungbean (Vigna radiata) intercropped with rice mixtures. We evaluated the effects of (1) rubber tree age, (2) rice cultivar and the combination of cultivars, and (3) rice and mungbean intercropping, on crop yield and the level of biotic constraints—disease severity, insect abundance, and weed biomass. We also illustrate the use of a split plot design in generalized linear mixed models for intercropping that could be useful for analysis of other intercropping experiments, and an analysis of the yield ranking of each cropping system across site-year combinations to evaluate mean relative performance and yield stability.

Materials and Methods

Experimental sites

The experimental sites were in the Arakan Valley Complex in the province of Cotabato in Central Mindanao, Philippines. Arakan has 28 villages, 10 of which are major rice producing villages. Its total land area is 69,432.79 ha with about 16,798 ha utilized for crop production. The landscape of Arakan is dominated by rolling hills, valleys, and mountain ranges.

Planting design

We evaluated the set of eight cropping treatments (Table 1), described in more detail below, in farmers’ fields for three seasons (2006–2008), with planting date details in Table S1. In 2006, two fields, one with 1-year-old rubber trees and one with 3-year-old rubber trees, were selected in each of three municipalities—Antipas, Arakan, and Pres. Roxas—to represent local systems. The experimental design in 2006 was a split-plot design with rubber tree age as the whole plot treatment (Fig. S1A). Each farm contained three complete blocks, with each block containing 10 m × 4 m subplots that were assigned to the eight intercropping treatments (Table 1; Fig. 1 and S1B).

Table 1 Experimental treatments planted in subplots between rows of rubber trees in farmers’ fields in Mindanao.

Treatment abbrev.	Subplot treatment between rubber trees (Applied to subplot with 10 rows)	Crops appearing in each treatment (rows)	
		Dinorado	UPL Ri-5	Mungbean	
Control	No crops planted				
Dinorado	10 rows rice cv. Dinorado (D)	10			
UPL Ri-5	10 rows rice cv. UPL Ri-5 (U)		10		
RM	Rice mixture: 2 rows D, 3 rows U, repeated twice	4	6		
0.5 MB	2 rows D, 3 rows U, 5 rows mungbeans (MB)	2	3	5	
0.8 MB	4 rows M, 2 rowsa U + D, 4 rows MB	1	1	8	
0.2 MB	2D, 2U, 2D, 2U, 2MB	4	4	2	
MB	10 rows mungbeans (MB)			10	
Notes.

a UPL Ri-5 and Dinorado in fractions of rows.

Due to water-logging in some municipalities, the experiments in 2007–2008 were moved. During these years, two-year experiments were conducted in three fields with 1- to 2-year-old rubber trees in the municipality of Arakan, in the villages of Doroluman, Naje, and Sabang. The experiment was established in 2007 on 17 July (Doroluman, with 1-year-old rubber), 12 June (Sabang with 1.5-year-old rubber), and 16 May (Naje with 2-year-old rubber) (Table S1). The experimental design in 2007–2008 was a split-plot design with the two years as repeated measures (Fig. S2). The whole plots were again individual farms, with farm effects considered a random effect. Each farm again contained three complete blocks, with the eight intercropping treatments (Table 1) applied to the subplots.

Figure 1 Planting rice and mungbean in an experimental site in Mindanao with three-year-old rubber.

Eight rice and mungbean treatment combinations (Table 1) were planted in plots between the rubber tree rows. Two rice cultivars were included in the design: Dinorado, a higher-value but lower-yielding cultivar, and UPL Ri-5, a higher-yielding but lower-value cultivar, along with a high-yielding mungbean variety, Pag-asa 7. Subplots were positioned between the rows of rubber trees. Each subplot was 10 m long and included 10 rows of rice and/or mungbean, with 0.4 m between rows and 0.2 m between hills for both rice and mungbean. The distance between the subplots and the base of the rubber trees was 1–1.5 m, and there was a 4 m border between the experimental plots to reduce interplot interference. In general pesticides were not used, with the exception of the Oreta one-year-old rubber farm in 2006. Hand weeding was implemented following standard farmer practices to support establishment of the crops.

Statistical analysis

The response variables evaluated were crop yield, land equivalent ratio, weed biomass, crop height, and the disease and pest ratings for each crop. The responses represent a range of different probability distributions for statistical analyses. Several response variables were approximately normally distributed, so these data were analyzed in a linear mixed model using the MIXED procedure in SAS (Version 9.2). The details of the ANOVAs are given in Tables S2 and S3, following the design structures in Figs. S1 and S2.

The same model structure was used for generalized linear mixed models for response variables that were not normally distributed, such as some disease ratings. The assumption of normality was tested using a Shapiro–Wilk test, and a Q-Q plot was evaluated where a heavy tail suggested use of a gamma distribution. Because of an upward skew in some data, the gamma distribution with log link function was used to analyze those responses using SAS Proc GLIMMIX. The ilink option was used for calculating the least square means. A Tukey–Kramer adjustment (at significance level 0.05) was used in multiple comparisons of performance within a crop across the different treatments in which that crop occurred. Weed dry weight was evaluated for each treatment (not evaluated separately for different crops). Weed weight was approximately normally distributed after square root transformation. We evaluated the effects of rubber tree age and cropping treatments (including the no-crop treatment) on weed weight in a linear mixed model. Ratings of disease severity were treated in analyses as approximately continuous.

Yield and economic value of intercrops: rice and mungbean

Both UPL Ri-5 and Dinorado have been preferred rice varieties of Arakan farmers for various reasons (RF Hondrade, pers. obs., 2006). Both mature in approximately 128 days. Their cooked grain quality is acceptable to consumers, while Dinorado has greater volume increase and better palatability. UPL Ri-5 is higher yielding (average yield approximately 2.5–3.5 t ha−1 under favorable conditions) compared to Dinorado (yield less than 2 t ha−1 up to 2.7 t ha−1 under the same management practices). However, milled Dinorado had a local market price of more than a dollar (in Philippine pesos, P45–50 kg−1, compared to UPL Ri-5 at P32–P35 kg−1). Higher yielding UPL Ri-5 helps ensure farmers’ household food security while the higher market value of Dinorado provides additional income to the farmer. The average yield of Dinorado in Arakan ranged from <1 to 1.5 t ha−1 while UPL Ri-5 produced 1 to 2 t ha−1. The national average yield of upland rice was <1 to 1 t ha−1.

Mungbean is a preferred legume of local farmers in Arakan, with a market price similar to Dinorado (RF Hondrade, pers. obs., 2006). Average yields under favorable growing conditions in the uplands ranged from 0.9 to 1.7 t ha−1. Farmers grow mungbean both for market and household food consumption.

In addition to the analysis of grain yield described above, yield was also used to calculate the corresponding land equivalent ratio (LER) for evaluating the rice mixture and the rice-mungbean intercrops in each subplot. The LER for each subplot was calculated using the following formula, illustrated for a two crop mixture. P1⋅Yield1in mixtureYield1in monoculture+P2⋅Yield2in mixtureYield2in monoculture

where Pi is the proportion of rows planted to Crop i in the subplot being considered (Table 1), and Yieldi is the yield of Crop i calculated based on the dry (air- or sun-dried) weight per row. The LER for the three crop mixtures (with Dinorado, UPL Ri-5, and mungbean) was calculated similarly.

The LER for each mixture is a measure of how well the mixture or intercrop yield compared to the monoculture yield. Pair-wise comparisons between the means of the LER for the four cropping treatments (Table 1) that included more than one type of crop were performed. A one-tailed t-test was performed to see if there was evidence that the four LER means were larger than 1.

Crop weed, disease, and insect evaluation

In each of the subplots, including the control, the weed biomass was weighed at the end of the season. Common diseases were evaluated visually by an experienced disease observer, as described below. A range of beneficial and pest arthropod species were also sampled within the subplots, including the rice bug (Leptocorisa acuta). The following diseases and insect pests were evaluated at crop maturity, where evaluation generally followed standard rating methods (INGER Genetic Resources Center, 1996). For rice panicle blast (caused by Magnaporthe oryzae), three categories of infection were recorded: no visible lesion, lesions on several pedicels or secondary branches, or lesions on few primary branches or the middle part of the panicle axis. Rice leaf blast (caused by Magnaporthe oryzae) was evaluated in 2007 and 2008, using a scale of ten possible categories of infection (INGER Genetic Resources Center, 1996). Rice brown spot (caused by Cochliobolus miyabeanus) was evaluated on a scale of 10 potential levels of severity. Pod rot of mungbean (caused by Gluconobacter sp.) was evaluated in 2008 based on percentage incidence, the percentage pods infected. Rice bug damage was evaluated in 2006 by damage category.

Analysis of relative yield ranking of treatments, and stability of yield rank across environments

To evaluate the relative yield and yield stability of the different cropping systems, we compared cropping system performance across the different environments, in an analysis similar to the Mundt (2002) analysis of wheat cultivar mixture performance. The analysis was performed as follows. The mean yield per row was computed for each treatment in each site for the three years. For each of the twelve site-years, the weighted mean yield across the seven cropping systems in Table 1 (excluding the control) was calculated, as an index of the quality of that site-year as an environment for crop production (“environment index”). For each site-year, the rank of each treatment mean yield was also computed (1 = lowest, 7 = highest), indicating the relative performance of each treatment in each site-year.

The performance of the intercropping systems was evaluated in two analyses, one based solely on yield and one based on yield weighted by economic value. For the first, we used a regression analysis to evaluate the relationship between the environment index (mean yield in each site-year) and the yield ranks for each treatment. The best intercropping system will have the highest mean yield rank and will maintain its yield rank across environments. In the regression analysis, a positive slope indicates higher performance rank in high yield environments, while a negative slope indicates higher performance rank in low yield environments. Lower P-values associated with the slope indicate stronger evidence that the performance rank of a cropping treatment changes with environment. A relatively small mean square error from the F test in the regression analysis indicates that the quality of the environment explains most of the variability in performance rank of a cropping system. Second, we performed the same type of analysis but with the crop yields weighted by their relative economic value. The economic value of Dinorado and mungbean was approximately equivalent, and 40% higher than the value of UPL Ri-5, based on typical valuation of these crops. Thus relative economic weights were 1 for UPL Ri-5, 1.4 for Dinorado, and 1.4 for mungbean, and the treatment mean yield per row was multiplied by these weights in the economic analysis.

Results

The responses for which the gamma distribution was used in SAS Proc GLIMMIX for 2006 were Dinorado and mungbean yield, disease levels for panicle blast and brown leaf spot, and rice bug levels. In 2007–2008 the gamma distribution was used in analysis of UPL Ri-5 and levels of brown spot, brown leaf spot, panicle blast, leaf blast, pod rot, and rice bug.

Rubber tree age effects

Rubber tree age did not have a significant effect on cropping system productivity (Table 2). Although the mean subplot yield in one-year-old rubber sites was 1.5 times that in three-year-old rubber sites, the variation among sites was very high (with yield 4.3 times higher in the highest yielding one-year-old rubber site compared to the lowest yield one-year-old rubber site).

Table 2 Treatment effects and results of an AOV for yield (grams/row) of mungbean and two rice cultivars (Dinorado and UPL Ri-5) from intercropping systems (Table 1) in rubber plantations in Mindanao.

Crops	2006 yield	2007–2008 yield	
	Effect	F-test P-values	Effect	F-test P-values	
Dinorado	Treatment	0.03	Treatment	0.51	
Rubber age	0.53	Year	0.21	
Trt*Rubber age	0.85	Trt*Year	0.73	
UPL Ri-5	Treatment	0.001	Treatment	0.88	
Rubber age	0.84	Year	0.15	
Trt*Rubber age	0.77	Trt*Year	0.07	
Mungbean	Treatment	0.45	Treatment	0.03	
Rubber age	0.53	Year	0.33	
Trt*Rubber age	0.43	Trt*Year	0.002	
Notes.

Bold P-values are significant at the 0.05 level.

Rice and mungbean yield responses to cropping system

Rice yields per row were greater in intercropping systems than in monoculture in some cases, although yields varied widely within and among years (Fig. 2 and Tables 2–5). There was a tendency for rice yields to be higher when rice made up a smaller proportion of the intercrop rows (Fig. 2 and Figs. S3). Dinorado yield responded significantly to intercropping treatments only in 2006 (p = 0.03), when Dinorado yield per area was significantly higher in 0.8 MB than in 0.2 MB (Tables 2 and 3). UPL Ri-5 yield also responded significantly to intercropping treatments only in 2006 (p = 0.001). UPL Ri-5 yield per area was significantly higher in 0.8 MB compared to the monoculture and RM in 2006 (Table 3). There were no significant treatment effects or significant mean differences in pairwise comparisons for the two rice varieties in 2007 and 2008 (Tables 3 and 2). As expected, UPL Ri-5 yield was generally higher than Dinorado yield.

Figure 2 Yield (g/row) of two rice cultivars (Dinorado and UPL Ri-5) and mungbean grown between rubber tree rows in a Mindanao plantation.

The rice and mungbean were grown in monoculture and in a set of mixture and intercropping treatments (Table 1). In the boxplots, the white bar indicates the median across all farms, the boundaries of the box indicate the 25th and 75th percentiles, the extent of the dotted lines indicate the minimum and maximum, and circles beyond these indicate more unusual values.

Table 3 Yield (grams/row) of mungbean and two rice cultivars (Dinorado and UPL Ri-5) for eight cropping treatments (Table 1), from intercropping systems in rubber plantations in Mindanao.

Crops	Treatment	2006 Yield	2007–2008 Yield	
		Mean	(SD)	Mean	(SD)	
Dinorado	Monoculture	487ab	195	697a	186	
RM	524ab	195	653a	186	
0.5 MB	626ab	195	1,075a	186	
0.8MB	762a	195	878a	186	
0.2 MB	470b	195	643a	186	
UPL Ri-5	Monoculture	651a	226	1,053a	185	
RM	605a	226	928a	185	
0.5 MB	790ab	226	1,044a	185	
0.8 MB	928b	226	1,252a	185	
0.2 MB	668ab	226	1,065a	185	
Mungbean	Monoculture	218a	75	367ab	31	
0.5 MB	178a	75	397ab	31	
0.8 MB	293a	75	348a	31	
0.2 MB	193a	75	430b	31	
Notes.

Treatments RM, 0.5 MB, 0.8 MB and 0.2 MB refer to the intercropping treatments in Table 1. Superscripts a, b: if treatments are marked by the same letters, then there is no significant difference in the pair-wise comparison. If the means have different letters, then there is a significant difference at the 0.05 level.

Table 4 The land equivalent ratio (LER) for eight cropping systems (Table 1) of mungbean and two rice cultivars (Dinorado and UPL Ri-5), from intercropping systems in rubber plantations in Mindanao.

Results are given for a t-test of whether the LER is greater than 1.

2006	2007–2008	Overall Mean	
Treatment and effects	Mean	(SD)	t-test P-value	Treatment and effects	Mean	(SD)	t-test P-value		
RM	1.53b	0.14	0.0007	RM	0.93a	0.08	0.19	1.23	
0.5 MB	1.13ab	0.14	0.18	0.5 MB	1.21b	0.08	0.009	1.17	
0.8 MB	0.99a	0.14	0.47	0.8 MB	1.12ab	0.08	0.08	1.05	
0.2 MB	1.23ab	0.14	0.06	0.2 MB	1.13ab	0.08	0.06	1.18	
Trt effect				Trt effect					
P-value	0.006	–	–	P-value	0.03	–	–	–	
Age effect				Year effect					
P-value	0.50	–	–	P-value	0.05	–	–	–	
Trt*Age				Trt*Year					
P-value	0.22	–	–	P-value	0.70	–	–	–	
Notes.

The Trt effect refers to the four treatments RM, 0.5 MB, 0.8 MB and 0.2 MB. Trt*Age is the four treatments and the rubber age interaction in year 2006. Trt*Year is the four treatments and year interaction in 2007–2008. Superscripts a, b: if the means contain the same letters, then there is no significant difference in the pair-wise comparison. If the means have different letters, then there is a significant difference at the 0.05 level. Bold p values are significant at the 0.05 level.

Table 5 Weed biomass in eight cropping systems (Table 1) of mungbean and two rice cultivars (Dinorado and UPL Ri-5), from intercropping systems in rubber plantations in Mindanao.

The square root transformation was used in analysis, where the original unit for weed biomass was g/m2.

	2006	2007–2008	
Treatment and effects	Mean	(SD)	Mean	(SD)	
Control	31.2a	2.5	24.6a	1.4	
Dinorado	14.5b	2.5	15.5b	1.4	
UPL Ri-5	10.8b	2.5	13.7bc	1.4	
RM	11.9b	2.5	15.1bcd	1.4	
0.5 MB	13.2b	2.5	13.9bcd	1.4	
0.8 MB	13.3b	2.5	13.1bcd	1.4	
0.2 MB	12.2b	2.5	12.6cd	1.4	
MB	14.7b	2.5	11.7c	1.4	
Trt effect					
P-value	<0.01	–	<0.01	–	
Age effect					
P-value	0.36	–	0.09	–	
Trt*Age					
P-value	0.78	–	<0.01	–	
Notes.

Eight intercropping treatments (Table 1) were compared. The effects of treatments with the same letter superscript are not significantly different. Bold P-values are significant at the 0.05 level.

Mungbean yield did not respond significantly to intercropping treatments in 2006 (Table 3). Mungbean also showed a tendency for higher yield when mungbean made up a smaller proportion of the intercrop rows (Fig. 2 and Fig. S3). There was a significant treatment effect for mungbean yield in 2007 and 2008 (p = 0.03; Table 2). There was a significant treatment × year interaction (p = 0.002). In 2007 and 2008, mungbean yield per area was significantly higher in 0.2 MB compared to 0.8 MB (Table 3). Mungbean data were missing from one farm in 2006 and there were also other cases of missing data that reduced the statistical power for mungbean comparisons. One component of variability in mungbean response was the observed 90% severity of pod rot in one site (P. Roxas) in 2006.

Land equivalence ratios

The estimated LER for the rice mixture varied with year (Fig. 3; Table 4). The intercrop LER estimate for 0.2 MB was significantly greater than 1 in all three years, and 0.5 MB and 0.8 MB were also significantly greater than 1 in two out of three years. The intercropping treatments differed significantly in their effect on LER in 2006 (p = 0.006; Table 4). The mean LER was significantly greater in RM and lower in 0.8 MB in 2006 (Table 4). Also, the LER for RM was significantly greater than one. In 2007 and 2008 there was a significant intercropping treatment effect on LER (p = 0.03). Among the pairwise comparisons, the 0.5 MB LER was significantly greater than the RM LER. Also the 0.5 MB LER mean was significantly greater than one, but the mean of RM was not significantly less than 1.

Figure 3 The Land Equivalent Ratio (LER) for a rice mixture and three intercropping systems (Table 1) in three years across all rubber plantations studied.

In the boxplots, the white bar indicates the median across all farms, the boundaries of the box indicate the 25th and 75th percentiles, the extent of the dotted lines indicate the minimum and maximum, and circles beyond these indicate more unusual values. These values were not weighted by the relative economic value of different crops.

Weed responses to cropping systems

Weeds commonly observed in the experiments were Ageratum conyzoides (Asteraceae), Borreria laevis (Rubiaceae), Calopogonium mucunoides (Fabaceae), Chromolaena odorata (Asteraceae), Ludwigia octovalvis (Onagraceae), Murdannia nudiflora (Commelinaceae), and Rottboellia cochinchinensis (Poaceae), where A. conyzoides was particularly abundant. As expected, all crop treatments resulted in lower weed biomass than the unplanted control (Table 5). The mungbean monoculture had the lowest weed biomass in 2007–2008, and was significantly lower than the weed biomass in the Dinorado monoculture and RM; however, in 2006 the mungbean monoculture did not significantly reduce weed biomass (Table 5), probably because of problems with mungbean establishment in that year. The intercropping treatment effects were significant in both 2006 and 2007–2008. There was also a significant treatment by year interaction in 2007–2008 (Table 5).

Disease and insect responses to cropping treatment

Disease and insect pests were present at low levels overall (Tables S10 and S12). Brown spot, panicle blast, leaf blast, and rice bug were common enough in at least some years to be evaluated for differential responses to the intercropping treatments. Brown spot (caused by Cochliobolus miyabeanus) was generally more severe in Dinorado than in UPL Ri-5, and there was evidence for decreased brown spot in Dinorado in the rice mixture and increased brown spot in Dinorado in the 0.8 MB intercrop in 2007–2008 (Table S12). Panicle blast (caused by Magnaporthe grisea) was generally more severe in UPL Ri-5 compared to Dinorado in 2006 (Table S10), but more severe in Dinorado than in UPL Ri-5 in 2007–2008 (Table S12). There was evidence that panicle blast was higher in Dinorado in the 0.2 MB intercrop and lower in the 0.5 MB intercrop in 2006 compared to other treatments (Table S10). Leaf blast could be evaluated in 2007–2008. Leaf blast severity was generally higher in Dinorado than in ULP Ri-5 (Table S12), but there were no significant cropping treatment effects (Table S13). In 2007–2008 there was a year effect on brown spot and leaf blast in UPL Ri-5 (Table S13). In 2006 there was an effect of rubber age on panicle blast in UPL Ri-5, where older rubber trees were associated with higher disease (Table S11). Pod rot of mungbean was observed at one site in P. Roxas in 2007–2008 where severity reached 90%, and the treatment effect was significant (Table S13).

The rice bug, Leptocorisa acuta, could be evaluated in 2006 and was generally more abundant in UPL Ri-5 than in Dinorado (Table S10). Rice bugs were significantly less common in Dinorado in all intercropping systems compared to Dinorado monoculture, but were more common in Dinorado in the rice mixture compared to the monoculture (Table S10). Other insect pests observed in some sites were green leafhoppers (Nephotettix spp.), brown planthoppers (Nilaparvata lugens), and grasshoppers. The most commonly observed beneficial insects were spiders, wasps, red ants, and lady bugs, but these were observed too infrequently to allow statistical comparison of treatment effects.

Yield rank and stability

In the analysis of yield rank and stability (Table 6), the monoculture UPL Ri-5 had the highest mean yield rank, with little evidence (P = 0.15) for a positive slope, and mean square error 1.95. The second highest mean yield rank was for the rice mixture (RM, 4 rows of Dinorado and 6 rows of UPL Ri-5), with a relatively high mean square error 2.53 indicating higher variability than observed for UPL Ri-5. The next highest ranked cropping systems were the 0.5 and 0.2 MB intercropping systems, with similar performance. Comparing all the treatments, the monoculture mungbean (MB) treatment had the lowest mean yield rank (1.9), a negative slope (−0.000377) and a relatively high mean square error (3.21) reflecting yield loss to water-logging problems at some sites, particularly in 2006.

Table 6 An analysis comparing the yield performance and stability of intercropping system treatments (Table 1) of mungbean and two rice cultivars (Dinorado and UPL Ri-5), in rubber plantations in Mindanao.

Regression analysis using the mean yield of each treatment for each site as predictor and yield ranks (1, lowest; 7, highest) of each treatment as response.

Treatment	Mean Rank	Slope	P-valuesa	MSEb	
UPL Ri-5	6.3	0.0025	0.15	1.95	
RM	5.0	0.0025	0.20	2.53	
0.5 MB	4.5	−0.0022	0.13	1.34	
0.2 MB	4.2	−0.0001	0.93	0.97	
Dinorado	3.8	0.0038	0.06	2.33	
0.8 MB	2.4	−0.0027	0.07	1.33	
MB	1.9	−0.0038	0.10	3.21	
Notes.

a Probability that slope is significantly different from zero based on F-test.

b Mean square error of the regression.

The yield rank and stability weighted by economic values (Table 7) showed similar trends. UPL Ri-5 still had the highest ranked performance, but among the weighted ranks there was stronger evidence (P = 0.07) for a positive slope, indicating that the relative performance of UPL Ri-5 was higher in environments with higher mean yield. The mean performance of the 0.5 MB intercropping system and the Dinorado monoculture was higher when weighted by economic values, tied for second highest rank. For the 0.5 MB system, there was not evidence for a non-zero slope, thus not evidence for a change in yield rank across environments (P = 0.3). For the Dinorado monoculture, there was evidence for a positive slope (P = 0.03), indicating Dinorado yield was less stable in poorer environments (Fig. S5). The Dinorado monoculture also had a somewhat higher MSE than the 0.5 MB intercropping system.

Table 7 An analysis comparing the yield performance (weighted by relative economic value) and stability of intercropping system treatments (Table 1) of mungbean and two rice cultivars (Dinorado and UPL Ri-5), in rubber plantations in Mindanao.

Regression analysis using the mean yield of each treatment for each site as predictor and yield ranks (1, lowest; 7, highest) of each treatment as response.

Treatment	Mean Rank	Slope	P valuesa	MSEb	
UPL Ri-5	5.3	0.0032	0.07	2.57	
0.5 MB	4.8	−0.0015	0.34	2.52	
Dinorado	4.8	0.0040	0.03	2.89	
0.2 MB	4.3	−0.0004	0.76	1.65	
RM	4.2	0.0008	0.64	2.71	
0.8 MB	2.6	−0.0027	0.05	1.54	
MB	2	−0.0033	0.11	3.85	
Notes.

a Probability that slope is significantly different from zero based on F-test.

b Mean square error of the regression.

Discussion

Crop production in rubber tree plantations appears to be a viable approach for increasing local food production where young rubber plantations are common, and potentially for increasing local household food security. As rubber trees age, the mean yield of crops intercropped with rubber may go down, but the difference between crop yield in one-year-old rubber and three-year-old rubber was small relative to the overall variability in yield. Farmers in the area of these experiments continued to plant rice, mungbean, peanut, or corn in zero- to two-year-old rubber, although they did not commonly use intercropping of crop species, and the practice of planting crop species between tree rows has also been adopted in palm oil plantations (RF Hondrade, pers. obs, 2008). Two rice varieties are commonly used, a higher yielding variety and either Dinorado or another traditional variety, depending on market demand (RF Hondrade, pers. obs., 2008).

Extension programs can help make farmers aware of the range of options for intercropping in tree crops like rubber. Mindanao had relatively higher rates of technological change in rice production than some other areas of the Philippines, with factors such as investment in infrastructure, farm mechanization, and adoption of modern rice varieties (Umetsu, Lekprichakul & Chakravorty, 2003). Planting rice with leguminous tree species has been proposed for upland rice production in eroded areas of northern Mindanao, with reports of increased UPL Ri-5 yield in eroded sites (MacLean et al., 2003). Farmers in Sri Lanka were more likely to adopt rubber intercropping if they had more extension contacts and higher education (Herath & Takeya, 2003). Intercropping with tea has been recommended to improve income in the pre-tapping phase for farmers in Sri Lanka, where intercropping in rubber can be an important part of household strategies with reported benefits both before and during rubber production (Rodrigo, Thenakoon & Stirling, 2001). Farmers were more likely to adopt this system if they had incomes above a minimal level, if their income was based solely on their farm, and if the majority of land was suitable for tea cultivation. Technical knowledge of how to intercrop tea and rubber was identified as a limiting factor in adoption (Iqbal, Ireland & Rodrigo, 2006).

In comparing the performance of the seven cropping system treatments (Table 1), a monoculture of UPL Ri-5 had the highest mean yield rank, with or without weighting for economic value per unit production. However there was some evidence that the performance of UPL Ri-5 was lower in poorer environments, where household food security may be more problematic. When considering yield weighted for economic value, the next highest mean yield rank was the tied 0.5 MB intercropping system and the Dinorado monoculture, where the 0.5 MB system appeared somewhat more stable (having a non-significant slope in response across environments and lower MSE). The poorer performance of some cropping systems was a result of poor mungbean establishment at some sites, which could potentially be improved through greater farmer experience in mungbean production. The tendency for crops to yield higher when they made up less than 50% of a cropping system suggests there may be more opportunities for developing useful intercropping systems.

Disease and pest levels were generally relatively low in all treatments, and these low levels present an interesting question in their own right. In fact this region of Mindanao has a reputation for having low disease pressure, where the reasons for this are not entirely clear. There have been outbreaks of bacterial blight and blast in Mindanao, which might be explained in part by widespread planting of a single rice cultivar (RF Hondrade, pers. comm., 2006). Part of the popularity of Dinorado may be due to its relative disease resistance, in addition to higher economic value. UPL Ri-5 produced higher yields than Dinorado, even when weighted for economic value. There was a tendency for UPL Ri-5 to have lower disease levels when in the mixture with the more resistant Dinorado, consistent with other reports for rice disease in mixtures (Meung et al., 2003; Zhu et al., 2000). There is the potential for mungbean to change the microenvironment for rice, but the difficulties in mungbean establishment may have made this harder to interpret in this experiment. Microclimate (canopy moisture) has probably played an important role in the success of some rice mixtures, where taller susceptible rice plants surrounded by shorter resistant plants experience less leaf surface moisture and so lower disease development (Zhu et al., 2005). Environmental differences among farms may alter the effects of crop mixtures due to altered competition, or to altered epidemic processes (Garrett et al., 2009). Weed species composition in upland rice is particularly variable compared to other rice production systems, and particularly challenging. Imperata control has been identified as an important component of management for Indonesian rubber (Grist & Menz, 1996). Potential management by planting rubber at high densities to compete with weeds presents a trade-off in that high-density rubber will make incorporation of food crops more challenging.

There are other possibilities for incorporating agroforestry systems, with the potential for a range of benefits including increased household food security and wider ecosystem services (Cheatham et al., 2009; Jose, 2009; Swift, Izac & van Noordwijk, 2004). For example, there is the potential to develop longer-term rubber intercropping systems, to buffer fluctuating rubber prices (Cramb et al., 2009; Nath, Inoue & De Zoysa, 2013), by altering rubber tree planting arrangements to allow greater resource availability for other crops (Rodrigo, Silva & Munasinghe, 2004), depending on the stage of development of the local systems (Barlow, 1997; Dressler & Pulhin, 2010) and the availability of other options for income (Dressler & Fabinyi, 2011; Langenberger et al., in press; Neyra-Cabatac, Pulhin & Cabanilla, 2012). In such a system, a managed understory can be integrated in place of additional trees. Upland rice presents a trade-off, because it has an important role in crop production for resource-poor farmers, but can present an important environmental cost if fragile ecosystems are converted for upland rice production. If lands are in plantation production, anyway, the environmental cost of adding upland rice is reduced. It will be useful to learn more about the probably complicated relationship between local rates of crop production in rubber plantations and household food security. Identifying more productive intercropping systems may help to promote useful implementation.

Supplemental Information

Supplemental Information 1 Supplementary Figures and Tables

Click here for additional data file.

Supplemental Information 2 SAS and R code, and data

Click here for additional data file.

We appreciate technical assistance from M Pinili, input from SG Elarde, G Lee, L Murray, and D Paranagama, and comments from PeerJ reviewers that led to improvements in the manuscript. We appreciate the contributions of the farmer-cooperators who maintained the field trials on their farms: Sim Arelolo, Rodolfo Bosque, Edwin Cainggoy, Wilson Espartero, Alex Oreta, and Arsenia Testado in 2006; W. Espartero, Rodrigo Pajanila, Romeo Pedroso, and A. Oreta in 2007 and 2008.

Additional Information and Declarations

Competing Interests

Author Contributions

Data Availability

The authors declare there are no competing interests.

Rosa Fe Hondrade and Casiana M. Vera Cruz conceived and designed the experiments, performed the experiments, wrote the paper, reviewed drafts of the paper.

Edwin Hondrade conceived and designed the experiments, performed the experiments.

Lianqing Zheng analyzed the data, wrote the paper, prepared figures and/or tables, reviewed drafts of the paper.

Francisco Elazegui performed the experiments, analyzed the data, wrote the paper, reviewed drafts of the paper.

Jo-Anne Lynne Joy E. Duque performed the experiments.

Christopher C. Mundt conceived and designed the experiments, wrote the paper, reviewed drafts of the paper.

Karen A. Garrett conceived and designed the experiments, analyzed the data, wrote the paper, reviewed drafts of the paper.

The following information was supplied regarding data availability:

The raw data has been supplied as a Supplementary File.

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
