# Peer review of "Cropping system diversification for food production in Mindanao rubber plantations: a rice cultivar mixture and rice intercropped with mungbean"

_PeerJ, doi:10.7717/peerj.2975_

## Round 0.1 · original submission · Major Revisions

The changes required are between major and minor - please address the referee comments, revise the existing ms, and return to me

Best wishes

Reviewer 1 ·

Basic reporting

This article evaluated the addition of bean and rice crops alone, and both combined as a mixed crop, established in the inter-row areas in rubber plantations in the Philippines. Although intercropping mixtures have been evaluated elsewhere before for different crops worldwide, the angle of mixing non-food crops with an alternative source of food for human consumption highlights that plant diversification in agroecosystems can promote food security without compromising the production of other valuable plant materials. In this sense, I really liked the idea behind this paper. However, two major concerns arose during the reviewing process: 1) Vague and sometimes not clear enough statistical analyses/results, and 2) Discussion at some extent was overly general and was not entirely based on the results presented. These points will be discussed below:

Experimental design

No comments

Validity of the findings

1) Vague statistics
I was a little confused about the statistical tests used in this article. Why using analysis of variance, (ANOVA presumably AOV in the text?), linear mixed models (LMM) and generalised mixed models (GLMM) instead of just using the GLMM? One of the advantages of the latter is that it can model different distributions, accounting for normality and heteroscedasticity without transforming the raw data (Bolker et al., 2009; Bolker, 2015). I used the term “Vague” here to characterise the use of “approximately normally distributed” in the main text (Lines 128, 129, 144), which is not clear. Unfortunately, the raw data set was not supplied (at least I was not able to use the data provided to perform those statistical analyses by my own). Therefore, further evaluation of the results presented here was not carried out.
Please check how to report statistical values after modelling, such as Fx,y=?; df=?; p=?. The interpretation of statistical analyses is sometimes not right. See line 292 and 302.
2) The authors did an interesting effort in communicating how this intercropping technique was adopted by Philippine farmers. However, this was never measured or reported in the current article. I suggest to the authors to look at (Gurr et al., 2016) for a similar pesticide/economic/social analysis. Also, (Khan et al., 2000) provided a very interesting approach in evaluating the adoption of novel agricultural techniques by farmers. Probably include other references in relation to this?
From lines 328 to 336 is not quite clear how these are connected with your work. Discussion needs to be written more concisely to better contextualise the results from this work.

Additional comments

I strongly recommend the authors to improve the English writing, as in some parts of the manuscript sentences are not connected between each other or unclear.
Discussion should be more focused on the actual measured results, rather than to account for something that was not evaluated in the paper (adoption of the techniques by farmers), despite that this adoption might be already applied on several farms around the studied landscape. If data on farmers’ surveys about the adoption of this technique is available, it should be included in the manuscript. If that is the case, I suggest revising the works of (Altieri, Funes-Monzote & Petersen, 2011), (Latour, 19999) and (Warner, 2007).
If the core idea of implementing a more diversified crop in non-food agricultural areas to promote food security in Philippine rubber plantations is currently applied in those areas, the article should properly measure and discuss them.
I encourage the authors to revise the manuscript, as the experimental split-plot design presented here can give an interesting statistical power to test the aims of this work.

Annotated reviews are not available for download in order to protect the identity of reviewers who chose to remain anonymous.

·

Basic reporting

The paper is very well written and generally clear. The citations are appropriate and the introduction provides a clear context for this important area of research.
Important clarification and streamlining:

1. Provide diagram of design and clarify LER calculation as discussed below under “Experimental Design.”
2. There are a lot of tables, and they are well-constructed with valuable information. However, given the low disease/pest pressure and lack of treatment effects on these responses, in would be efficient to move Tables 6-9 to the Supplemental Tables & Figures category. The textual description of the results should remain.

Minor points:

Abstract: Last sentence should be “adopted” rather than “adapted?”
L56: Define “land expectation value.”
L59: A couple quantitative examples of improvements would be good.
L80: Do rubber growers stop intercropping once the trees begin to be tapped? Is that an option or not something the authors envision?
L86-89: Put what was evaluated first and response variables second.
L148-162: This may be more appropriate in the Introduction.
L177-192: More details here on intensity of sampling for pests and diseases and rating systems used. References to established systems as given for rice leaf blast is sufficient, but only indicating “evaluated in 2006 by damage category” is not. Indicate any treatments for weeds—I infer that there were none.

Experimental design

The research question is within the scope of the journal, well-defined, highly relevant, and meaningful. The approach is straightforward though the nested design and multiple sites makes description challenging. Assuming I’m interpreting the text properly, the design is adequate and robust under the difficult circumstances of on-farm trials and events such as flooding.

Important points:

1. Because of the split-plot nature of the design and the change of sites, the explanation in the methods would benefit from a diagrammatic representation of the relationship between sites, farms, tree age, whole plots, and subplots. Perhaps a map of Mindanao with farm locations should be part of this, but the main thrust should be to clarify the hierarchy of experimental elements, and the replication.
2. Indicate the size of the plots. How long were the 10-row units, was one such unit considered one subplot or did the subplot traverse multiple tree rows? What constituted the buffer between the plots—the distance between gaps and size of the border plant areas? This is particularly relevant to disease and insect movement.
3. The calculation of LER is presented but it is unclear how it could be done without a monocrop of mung bean as one of the treatments. It would seem that the appropriate LER calculation for the rice mixture would be with the Dinorado and UPL Ri-5 plots as the monocrops and the RM as the “intercrop;” is this the case? The various actual interspecific mixtures should calculate LER using the RM as one monocrop and a mung bean monocrop as the other, but no such treatment is described.

Validity of the findings

The data is robust and statistically sound, and the conclusions well stated. The use of yield rank regressions is an excellent approach to evaluating the results. The appearance of outliers in Figs. 2 and 3 however should be explained in the text; i.e., how they were determined and the number of them relative to the number of samples for each treatment.

Additional comments

This well-designed and -reported research addresses an important issue for growers in Mindanao, that of balancing cash crops with food security, as well as taking advantage of the potential benefits of intercropping and reinforcing resilience of cropping systems. It can be extended to other systems and locations and adds to the growing empirical base for intercropping efficacy.

---

## Round 0.2 · accepted · Accept

Dear authors

You have worked hard on addressing referee comments

Many of these were of the type 'could do' or 'might be a good idea to' etc and in many cases you have done what's possible given the limitations of the work and available data.